# Small RNA in sperm–Paternal contributions to human embryo development

Signe Isacson [1], Kajsa Karlsson [1,2], Stefan Zalavary [2], Anna Asratian [1], Unn Kugelberg[1], Susanne Liffner[2] & Anita Öst [1] ✉

Sperm not only delivers the paternal genome to the oocyte but also regulatory small RNA (sRNA). However, the role of sRNA in fertilisation and human embryo development remains poorly understood. Here, couples undergoing IVF are recruited, and sperm sRNA analysed to investigate their role in IVF treatment. Differential expression of mitochondrial sRNA and Y-RNA are observed in relation to sperm concentration. For fertilisation rate, sRNA from a single locus are significantly changed. Expression of microRNA (miRNA) and ribosomal sRNA correlates positively and negatively, respectively, to high-quality embryos. Notably, the top miRNA have an area under ROC of >0.8. Predicted targets of these miRNA are relevant for development, suggesting a role for sperm-borne miRNA in embryo development. In conclusion, sperm-borne sRNA are biomarkers for sperm concentration and embryo quality in IVF. These findings may contribute to clinical strategies improving embryo quality, lowering costs and reducing the need for additional treatment cycles.

Infertility is increasing globally, affecting one in six couples of reproductive age[1]. To address this issue, researchers have developed assisted reproductive technologies, such as In Vitro Fertilisation (IVF), resulting in about 12 million babies being born[2]. Despite being well-established, IVF is only successful in ~30% of cases[3]. One major limiting factor in modern IVF treatments is embryo development. IVF clinics often aim to have 30% of fertilised oocytes develop into high-quality embryos, although this success rate is not reached consistently[4]. Often, couples need several rounds of treatments to receive a baby[5]. In this context, the potential for male contributing factors to increase IVF success has remained unexplored.

Accumulating evidence suggests that the sperm delivers not only genetic material but also small RNA (sRNA) to the oocyte. By definition, sRNA are short RNA molecules ranging from tens to a few hundreds of nucleotides. sRNA include microRNA (miRNA), originally discovered for their regulatory role in embryogenesis, as well as tRNA-derived fragments (tsRNA), ribosomal RNA-derived fragments (rsRNA), mitochondrial-derived RNA (mitosRNA), Piwi-interacting RNA (piRNA), and fragments from ribonucleoprotein associated RNA (ribonucleoprotein associated sRNA) such as Y-RNA and small nucleolar RNA[6].

Here, we show that by analysing sRNA in sperm samples retrieved during ongoing IVF treatment, we find subpopulations of sRNA to be biomarkers for sperm concentration, fertilisation rate, and embryo quality. Our work highlights the paternal role in IVF success and establishes a foundation for a new era in reproductive medicine.

## Results

### Characteristics of the population studied

Couples undergoing IVF were recruited between the 18 November 2022 and 8 June 2023 at the Centre of Reproductive Medicine at University Hospital in Linköping, Sweden. Of the 84 couples assessed for eligibility, 72 were recruited and pseudonymized. One couple had an inadequate sperm count for sRNA sequencing and was excluded after sample collection. Two couples did not retrieve any oocytes and were excluded from further analysis. One couple provided two samples because they underwent two IVF treatments; both samples were included. Thus, 70 treatments from 69 couples were studied, with sperm samples sequenced from each of these treatments (Fig. 1). The median age of the men was 34 years, and their median BMI was 27. The median age of the women was 33, and their median BMI was 25. Median

[1]Department of Biomedical and Clinical Sciences, Linköping University, Linköping, Sweden. [2]Division of Obstetrics and Gynaecology, Linköping University, Linköping, Sweden. ✉e-mail: anita.ost@liu.se

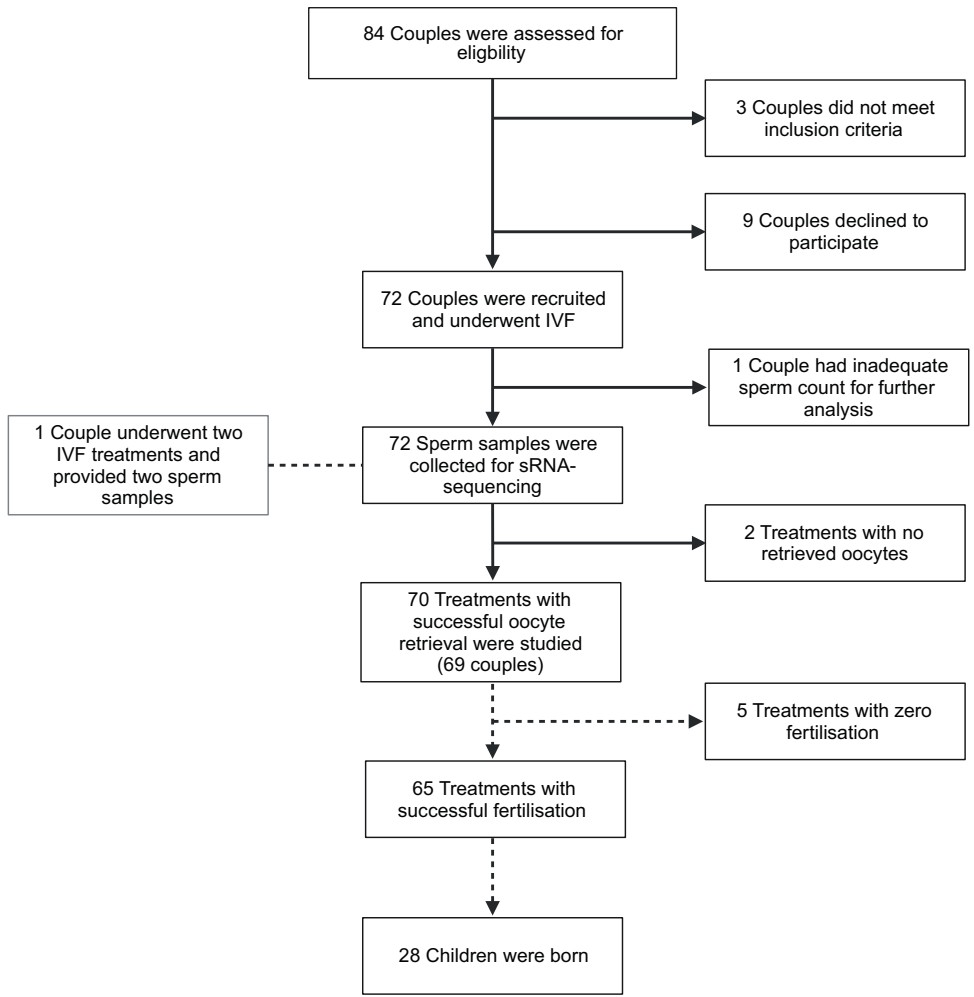

**Fig. 1 | Study flow chart.** IVF In vitro fertilisation. sRNA small RNA.

sperm concentration and progressive motility were 28 million sperm/mL and 50%, respectively (Table 1).

## Sperm-borne sRNA is differentially expressed in IVF sperm samples

To investigate whether there are molecular markers capable of predicting IVF outcome, leftover sperm was collected and sequenced. In total, 70 samples from 69 couples were sequenced for sRNA. Analysis of this data confirms previous findings that sperm contains a large and diverse repertoire of sRNA[7,8] (Supplementary Data 1, and Supplementary Fig. 1).

We performed differential expression analysis on sequenced sRNA and identified sRNA differing by sperm concentration, fertility rate, and rate of high-quality embryos. Specifically, differential expression analysis comparing samples with high (>16 million sperm/mL) and low (≤16 million sperm/mL) sperm concentration (Supplementary Data 2) identified 563 (1.89%) significantly upregulated sRNA in samples of high concentration (Fig. 2a, Supplementary Data 3). Conversely, 640 (2.15%) sRNA were significantly downregulated. Of the identified upregulated sRNA, 72% were mitosRNA; of the downregulated sRNA, 48% were ribonucleoprotein associated sRNA (Fig. 2a). As sperm concentration and motility are used to determine sperm quality, we performed differential expression analysis of these two parameters. We identified a great overlap between significant sRNA in sperm concentration and sperm motility (≥5 million progressively motile sperm as high and <5 million sperm as low motility), where 48% of all differentially expressed

sRNA were identical between the two parameters (Supplementary Fig. 2a–c).

Next, comparing sRNA from sperm associated with high (≥70%) and low (<70%) fertilisation rates (Supplementary Data 4), we identified 34 sequences (0.11%) to be downregulated in samples of high fertilisation rate (Fig. 2b, and Supplementary Data 5). Of these, 39% were piRNA, 34% unannotated, and 27% tsRNA (Fig. 2b). We found that no sRNA were significantly associated with high fertilisation rate. Lastly, when comparing sperm producing high (≥20%) respectively low (<20%) rates of high-quality embryos (Supplementary Data 6), 60 sRNA (0.20%) were upregulated and 104 sRNA (0.35%) were downregulated (Fig. 2c; and Supplementary Data 7). Upregulated sRNA consisted of 66% miRNA, and downregulated sRNA consisted of 73% rsRNA (Fig. 2c). Thus, differential expression analysis of sperm-borne sRNA identified unique RNA profiles associated with sperm concentration, fertilisation, and embryo quality.

## MitosRNA and Y-RNA are important for sperm concentration and motility

Next, we investigated sRNA differentially expressed in sperm concentration. Therefore, we mapped differentially expressed sRNA to their genes of origin and will hereafter name sRNA according to their annotation. Since the most prominent sRNA biotype upregulated in sperm concentration originated from the mitochondrial genome (Fig. 2a), we investigated these mitosRNA further. All upregulated mitosRNA originated from mitochondrial tRNA genes (Supplementary

**Table 1 | Demographic and clinical characteristics of the couples**

| | N | Median | Minimum | Maximum |
|---|---|---|---|---|
| Age male (years) | 69 | 34 | 25 | 50 |
| Age female (years) | 69 | 33 | 26 | 39 |
| BMI male (kg m²) | 69 | 26.5 | 18.4 | 48.4 |
| BMI female (kg m²) | 69 | 24.6 | 18 | 33.9 |
| Sperm concentration before gradient (million/mL) | 70 | 27.8 | 4 | 100 |
| Sperm concentration progressively motile sperm before gradient (million/mL) | 70 | 15 | 2 | 50 |
| Progressively motile sperm before gradient (%) | 70 | 50 | 21 | 84 |
| Motile sperm before gradient (%) | 70 | 55 | 26 | 84 |
| Progressively motile sperm after gradient (million) | 70 | 10.5 | 0.5 | 60 |
| Progressively motile sperm after gradient (million/mL) | 70 | 10.5 | 1 | 60 |
| Oocytes (n) | 70 | 8 | 1 | 19 |
| Total dose hormone (IU) | 69 | 2050 | 875 | 5363 |
| FSH:oocyte ratio | 69 | 275 | 75 | 5363 |
| Fertilised oocytes (n) | 70 | 4.5 | 0 | 15 |
| Fertilisation rate (%) | 70 | 67 | 0 | 100 |
| High-quality embryos (n) | 65 | 1 | 0 | 6 |
| Rate of high-quality embryos (%) | 65 | 33 | 0 | 100 |
| Live birth (n) | 28 | | | |
| Gestational age (days) | 28 | 277 | 245 | 292 |
| Birthweight (grams) | 28 | 3430 | 2770 | 4420 |

*Note. BMI* body mass index. Gradient is referring to the PureSperm Gradient for in vitro fertilisation sperm preparation (see "Methods section"). *IU* international units. *FSH* follicle-stimulating hormone. FSH:oocyte ratio = Total dose hormone (IU) / n oocytes. Fertilisation rate = n fertilised oocytes/n oocytes retrieved*100. High-quality embryos are defined in the "Method section". Rate of high-quality embryos = n high-quality embryos/ n fertilised oocytes*100. The couple with two treatments only contributes once to age and BMI. One female received hormonal treatment dosed in micrograms and not IU, therefore she is excluded from the parameters Total dose hormone (IU) and FSH:oocyte ratio.

Fig. 3). Of these genes, the three most significant were MT-TQ-Glu, MT-TH-His, and MT-TS1-Ser1 (Fig. 3a); MT-TS1-Ser1 was the most significant. Further examination with linear regression of sperm concentration to levels of sRNA mapping to MT-TS1-Ser1 showed a positive significant correlation ($R^2 = 0.208$, $P \le 0.0001$) (Fig. 3b). Additionally, the potential of MT-TS1-Ser1 to differentiate a low or high sperm concentration was assessed by performing a receiver operating characteristic (ROC) analysis, where the area under curve (AUC) was 0.891 (Fig. 3c).

In contrast, downregulated sRNA was mostly ribonucleoprotein associated sRNA (Supplementary Fig. 3b). The top three most significant all mapped to Y-RNA (Fig. 3d). Of these, sRNA mapping to RNY4 was the most significantly changed. Linear regression of sperm concentration and sRNA mapping to RNY4 showed a significant negative correlation ($R^2 = 0.238$, $P \le 0.0001$, Fig. 3e). The AUC was 0.845 in this ROC analysis (Fig. 3f). Thus, we identified mitosRNA and ribonucleoprotein associated sRNA from Y-RNA as biomarkers that differentiate between sperm samples with high or low sperm concentrations.

## Sperm's ability to fertilise oocytes relate to sRNA from a specific locus

We identified 34 sequences of sperm-borne sRNA to be associated with low fertilisation rate (Fig. 2b, Supplementary Data 5). A closer analysis of these sequences revealed that they all originate from the same genomic region (Fig. 4a). Sequences from this region can be annotated as both tRNA and piRNA depending on database. It is therefore not possible to precisely classify the differentially expressed sequences found here. Comparing the sum of normalised sequences in samples with high or low fertilisation rate did not show a statistically significant difference (Fig. 4b, c). In addition, ROC analysis showed an AUC of 0.5817 (Fig. 4d). In all, our data suggests that sequences from this specified region may be relevant for the sperm's capability of fertilisation, but more data would be needed to strengthen this finding.

## Embryo quality is affected by sperm-borne miRNA

Concerning the rate of high-quality embryos, miRNA was the most prominent biotype of upregulated sRNA. Specifically, 16 miRNA were found (Supplementary Fig. 4a); the three most statistically significant were hsa-let-7g, hsa-miR-30d, and hsa-miR-320b/a (Fig. 5a, and Supplementary Data 8). Gene Ontology (GO) term analysis of the predicted targets for these three miRNAs showed several biological processes to be involved. Interestingly, nine of the top GO terms were related to embryogenesis, development, or cell proliferation (Supplementary Fig. 5; Supplementary Data 9). This finding suggests that these miRNA play important roles in developmental processes. Further investigation of the miRNA revealed that hsa-let-7g had the lowest $P$ value ($P = 0.000000005.4$, Fig. 5a), making it an interesting candidate for further investigation. Comparing hsa-let-7g expression against the rate of high-quality embryos showed no significant regression ($R^2 = 0.001$, $p = 0.79$, Fig. 5b). However, exclusion of one sample with an expression 12 times above mean, resulted in a significant regression (Supplementary Fig. 4c). Moreover, an AUC of 0.812 for the ROC curve (Fig. 5c) supports the notion that hsa-let-7g affects embryo quality outcomes. Additionally, we studied linear regression and ROC analysis for hsa-miR-30d (Supplementary Fig. 4d) to address the sensitivity of miRNA for embryo quality. Regression was significantly positive ($R^2 = 0.065$, $p = 0.04$), and AUC was 0.712, suggesting that these miRNA are robust biomarkers of the ability of the sperm to produce high-quality embryos.

Additionally, downregulated sRNA were identified as mostly rsRNA. RsRNA originating from ribosomal genes 28S, 5S, 5.8S, and 12S were associated with sperm giving rise to low rates of high-quality embryos (Fig. 5d; and Supplementary Fig. 4b). Of these, rsRNA from the 28S gene were the most significantly changed between the high and low rate of high-quality embryos. Linear regression against sRNA mapping to 28S rRNA and rate of high-quality embryos showed a significant negative correlation ($R^2 = 0.084$, $P = 0.02$), and AUC was 0.79 (Fig. 5e, f). We found that miRNAs and sRNA from 28S were sperm-borne biomarkers that predict embryo quality in IVF treatment.

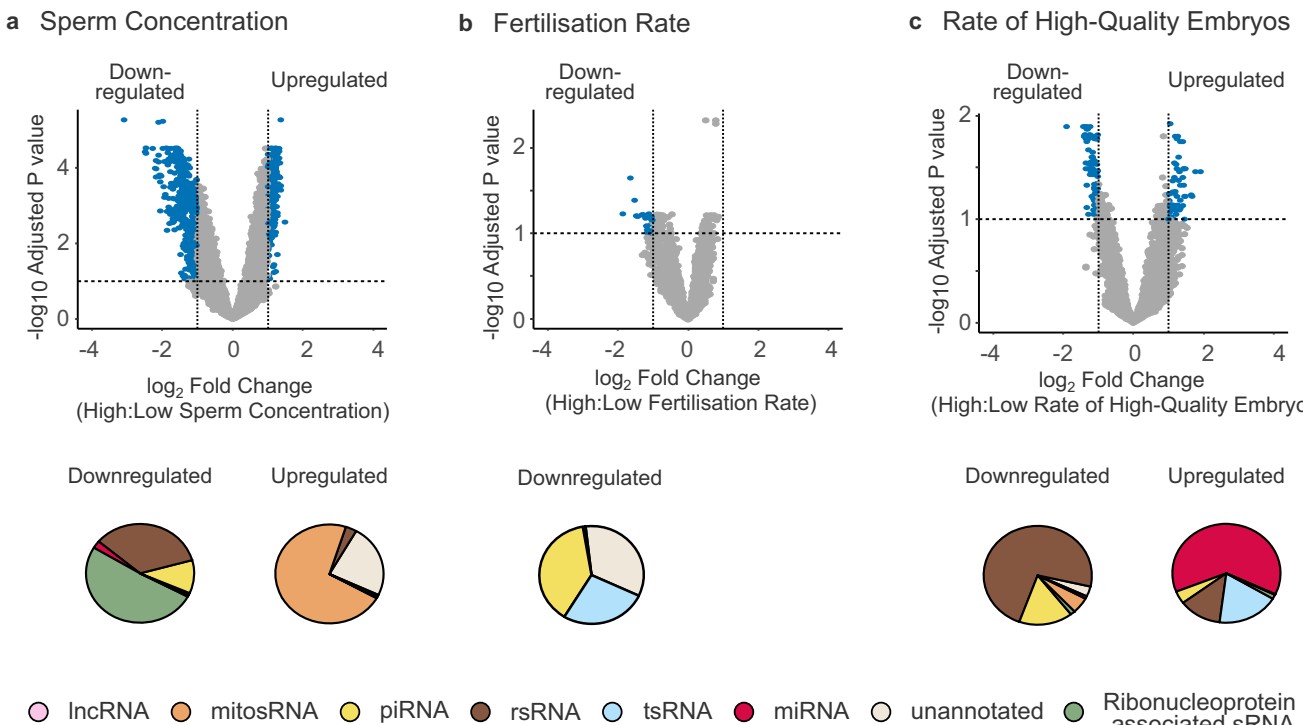

**Fig. 2 | Differential expression analysis of sRNA in sperm concentration, fertilisation rate and rate of high-quality embryos. a** Comparison of high and low sperm concentration (>/≤16 million sperm/mL). Pie charts represent up and downregulated sequences respectively, where the percentage of normalised sequences per biotype is shown. **b** Comparison of high and low fertilisation rate (≥/< 70%). Pie charts represent up and downregulated sequences respectively, where the percentage of normalised sequences per biotype is shown. **c** Comparison of high and low rate of high-quality embryos (≥/< 20%). Pie charts represent up and downregulated sequences respectively, where the percentage of normalised sequences per biotype is shown. Pink=long noncoding RNA (lncRNA), orange=mitochondrial small RNA (mitosRNA),

yellow = piwi-interacting RNA (piRNA), brown= ribosomal RNA derived fragments (rsRNA), blue=transfer RNA derived fragments (tsRNA), red=microRNA (miRNA), light beige=unannotated, green=ribonucleoprotein associated sRNA. Cpm=counts per million. Ribonucleoproteins include SRPs, vault-RNA, snRNA, snoRNA, scaRNA and Y-RNA. Upregulated are sequences with an adjusted *P* value of <0.1 and a log2 fold change above 1, whereas downregulated are sequences with an adjusted *P* value of <0.1 and a log2 fold change below -1. Adjusted *P* values and log2 fold changes shown in **a**, **b** and **c** were all produced by differential expression analysis with DESeq2 as described in the "Methods section". Up and downregulated RNA are marked blue in volcano plots.

## Chance of live birth and size for gestational age may be influenced by sperm sRNA

From the 65 treatments with successful fertilisation, 28 children were born (Fig. 1). To investigate whether there are molecular markers capable of predicting live birth, gestational age, and size for gestational age, we performed differential expression of sperm sRNA and these parameters (Supplementary Fig. 6a–c, and Supplementary Data 10). We identified a few sRNA to be upregulated in sperm associated with live birth (Supplementary Fig. 6a). These sequences originated from the snRNA U6 (Supplementary Fig. 6d). Regarding gestational age, there was no association with sperm-borne sRNA (Supplementary Fig. 6b). Next, looking at size for gestational age, we identified a subset of tsRNA (Supplementary Fig. 6c). These sequences mapped to four tRNA, Arg-CCT-3−1, iMet-CAT−1−1, Glu-TTC−14−1 and Lys-CTT-2−1 (Supplementary Fig. 6e–h, left panels). Sequences aligning to the same tRNA was summed and plotted as a function of size for gestational age. Three of these tRNA showed a negative correlation to size for gestational age ($R^2$ between 0.05811 and 0.1879), suggesting that these sequences relate to foetal growth (Supplementary Fig. 6e–h, middle panels). When grouping the samples in sperm giving rise to babies small for gestational age (SGA), average for gestational age (AGA), large for gestational age (LGA) and performing a non-parametric Kruskal-Wallis, there was no significant difference between the groups (Supplementary Fig. 6e–h, right panels). While the negative correlation between tsRNA and size for gestational age is an interesting observation, the few data points in SGA ($n = 2$) and LGA ($n = 5$), indicate that more data is needed to support this observation.

## Discussion

We provide evidence that sperm-borne sRNA are biomarkers for sperm quality, fertility rate, as well as embryo quality in an IVF setting. Our data reveal specific subpopulations that are relevant throughout IVF treatment. These findings provide intriguing new insights into the contributions of sperm in embryo development.

　　We identified mitosRNA to be markers of sperm quality in couples undergoing IVF (Fig. 3a; and Supplementary Fig. 3). In a previous study, we found that sperm-borne mitosRNA is affected by a short-term diet intervention[7] and directly related to human sperm quality[9]. Thus, there is accumulating evidence that mitosRNA are markers for high sperm concentration and motility. A recent investigation has shown that sperm-borne mitosRNA in mice, as in humans, are sensitive to a short-term diet intervention, suggesting that there are evolutionary conserved mechanisms in nutrient sensing capabilities[10]. Moreover, these sperm-borne mitosRNA appear capable of programming the metabolism in the developing embryo, suggesting that even though the mitochondrial genome is exclusively inherited from the mother, the father transmits mitochondrial regulation via the sperm. Further prospective studies, such as IVF cohorts, are needed to investigate whether a similar mechanism is involved in humans. Notably, we found that high levels of mitosRNA correlate with sperm concentration but not with fertilisation rate and embryo quality. This finding suggests that in an IVF setting sperm-borne mitosRNA are not important for embryonic development or fertilisation. However, sperm-borne mitosRNA may have functional relevance in in vivo fertilisation, where progressive motility in sperm is required to fertilise the egg.

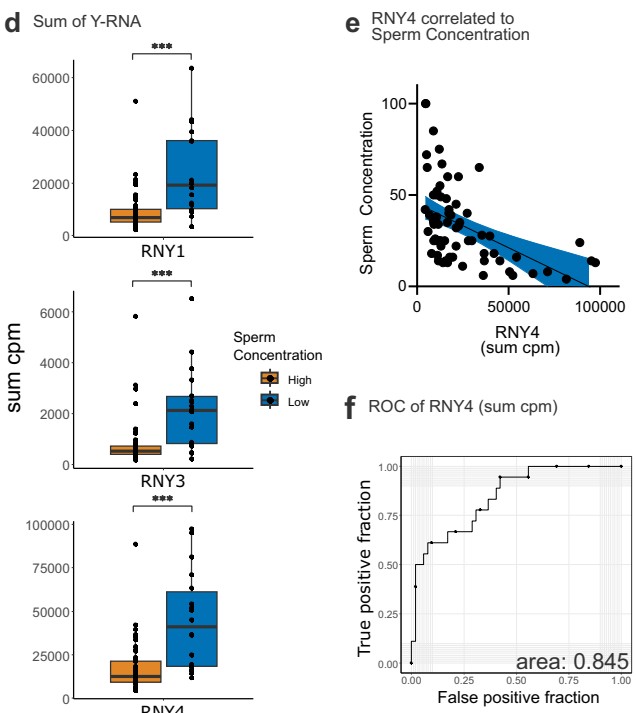

**Fig. 3 | sRNA differs between high and low sperm concentration samples.**
**a** MitosRNA with highest significance between high and low sperm concentration
(>/≤16 million sperm/mL). MT-TH-His $P$ val=0.000024, MT-TQ-Glu $P$ val=0.000054,
MT-TS1-Ser1 $P$ val=0.0000009. $P$ values were produced by Wilcoxon nonparametric
test (two-sided). **b** Linear regression of MT-TS1-Ser1 comparing sum cpm and sperm
concentration ($R^2$ = 0.208, $P$ value = <0.0001, equation= Y = 0,01862*X + 15,83). **c** ROC
of sum cpm of MT-TS1-Ser1 to differentiate between high and low sperm concentra-
tion. **d** Ribonucleoprotein associated sRNA with highest significance between high and
low sperm concentration. RNY1 $P$ val=0.000045, RNY3 p-val=0.00009, RNY4 p-
val=0.000015. $P$ values were produced by Wilcoxon nonparametric test (two-sided).
**e** Linear regression comparing RNY4 and sperm concentration ($R^2$ = 0.238, $P$ value =
<0.0001, equation= Y = −0,0004823*X + 45,23). **f** ROC of sum cpm of RNY4 to dif-
ferentiate between high and low sperm concentration. Orange=group of high sperm
concentration, blue=group of low sperm concentration. Data is presented as sum of
normalised sequences (cpm) for each sperm sample by indicated genomic origin. $P$
values for Wilcoxon nonparametric t-test, *≤0.05, **≤0.001, ***≤0.0001. High sperm
concentration $n$ = 51, low sperm concentration $n$ = 18. Line in box plot represents the
median, hinges show the first and third quartiles and whiskers extend to the largest
value unless values are above 1.5 times the inter-quartile range. In linear regression,
coloured area represents 95% confidence interval.

Examining the role of sperm-borne sRNA in embryo development,
we found 16 miRNA to be relevant (Supplementary Data 7, 8, and Sup-
plementary Fig. 4). GO term analysis of predicted targets from the top
three miRNA suggests a role for these miRNA in embryogenesis, devel-
opment, and cell proliferation (Supplementary Fig. 5, Supplementary
Data 9). Previous studies have reported that several sperm-borne miRNA
are important for male reproductive health[11–16], embryo quality[17,18], as
well as likelihood of live birth[19]. In a recent meta-analysis, Joshi et al.[20]
concluded that three miRNA are relevant biomarkers for male infertility.
Here, we found one of these to be a predictor of low rates of high-quality
embryos, supporting the relevance of this miRNA for embryo quality
(Supplementary Data 11). Similarly, Hamilton et al.[21] found sperm-borne
miRNA to be related to the rate of blastocyst formation. Of these, we
identified miR-200b and let-7a to be associated with low sperm con-
centration and miR-200b to be associated with low fertility rates.

Of the 16 miRNA identified in this study, three are members of the
let-7 family (Supplementary Data 8). This miRNA family is well-
established as a conserved family that is crucial for diverse aspects of
embryogenesis. Previous work suggests that let-7 regulates embryonic
and postnatal growth in mice via the insulin-PI3K-mTOR pathway[22].
Huang et al.[23] found that rodent sperm-borne let-7 dictates the meta-
bolic phenotype in offspring. Furthermore, they found that weight loss
in humans reduces levels of let-7d/e in sperm. Thus, our results com-
bined with mentioned findings support a functional role for sperm-
borne miRNA. From our data, however, we cannot infer whether the

level of miRNA in human sperm is causal to embryo quality or is a
consequence of other processes in sperm. In mice, it is clear that
sperm-borne miRNA are pivotal for preimplantation embryonic
development[24]. In humans, however, functional validation studies
would be required to provide further evidence. It is evident that sperm
miRNA, such as let-7g and mir-30d, are promising biomarkers that
could predict whether a sperm sample will result in high-quality
embryos in IVF treatment (Fig. 5a, c, and Supplementary Fig. 4c, d).
Notably, the let-7 family holds prognostic value in other contexts. In
terms of fertility, let-7c is important for blastocyst formation[25], preg-
nancy loss[26], and azoospermia[27] and let-7i is relevant for PCOS[28], IVF
success rate[29], endometriosis[30], and male subfertility[31]. Lastly, let-7g
has been associated with ovarian response to hormonal stimuli[29] and
foetal alcohol spectrum disorders[32].

In addition to mitosRNA and miRNA, we identified several other
relevant subtypes of sRNA for IVF. We found that sRNA originating
from Y-RNA – particularly RNY1, RNY3, and RNY4—are associated
with low-quality sperm. While sRNA from Y-RNA has previously been
reported to be present in human sperm[33], we found that sRNA from
Y-RNA is also relevant for male reproductive health. Full-length Y-
RNA are components of the Ro60 ribonucleoprotein particle,
involved in DNA replication and RNA quality control[34]. In addition,
our analysis supports the relevance of sperm-borne rsRNA for
embryo quality, where specific sequences of 28S rRNA exhibit a
lower expression in sperm samples producing high-quality embryos

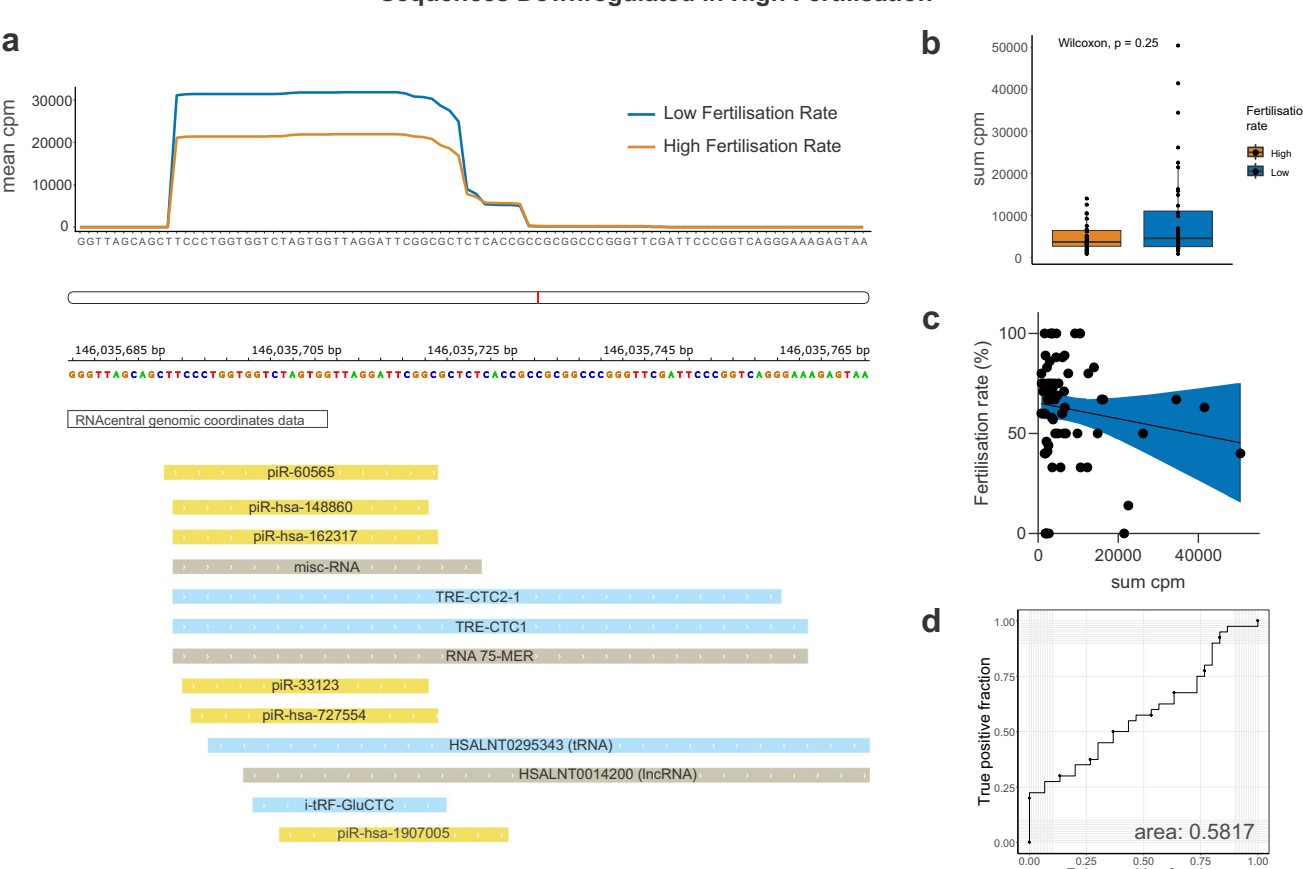

**Fig. 4 | Sperm sRNA differs between samples of high and low fertilisation rate.**
**a** Mean cpm of high and low fertilisation rate (≥/< 70%) of differentially expressed sequences from Fig. 2b (upper panel). Genomic region of chromosome 1, 146,035,680−146,035,770 from reference GRCh38. Annotation was extracted from IGV with RNAcentral genomic coordinates data version 24.0 (lower panel). **b** Sum cpm of all differentially expressed sequences in high and low fertilisation rate. *P* values as shown in figure were produced by Wilcoxon nonparametric test (two-sided). **c** Linear regression comparing sum cpm of differentially expressed sequences and fertilisation rate (R² = 0.01918, P value = 0.2529, equation= Y = −0,0003943*X + 65,25). **d** ROC of sum cpm of all differentially expressed sequences to differentiate between low and high fertilisation rate. Orange=group of high fertilisation rate, blue=group of low fertilisation rate. Data is presented as sum of normalised sequences (cpm) for each sperm sample by indicated genomic origin. High fertilisation rate n = 29, low fertilisation rate n = 40. Line in box plot represents the median, hinges show the first and third quartiles and whiskers extend to the largest value unless values are above 1.5 times the inter-quartile range. In linear regression, coloured area represents 95% confidence interval.

(Fig. 5d, f). In mice, sperm-borne rsRNA have been found to share biogenesis pathways with tsRNA via Dnmt2[35] as well as being phthalate-sensitive[36], but so far their roles in fertility have not been evaluated. Regarding fertilisation rate, we find sequences of piRNA or tsRNA originating from the same locus to be differentially expressed, suggesting a potential intriguing role for this locus in male fertility. We also find other sperm-borne tsRNA, namely from Arg-CCT-3−1, iMet-CAT-1-1, Glu-TTC-14-1 and Lys-CTT-2-1 to have a trend of negative correlation with size for gestational age. In mouse, sperm-borne tsRNA has been related to offspring body weight[37]. However, in a human setting, these findings need to be further explored in larger cohorts since foetal growth as well as fertilisation is heavily dependent on female factors before and after conception. Nonetheless, Oluwayiose et al.[38] identified two sperm-borne piRNAs to be relevant for the chance of live birth. Additionally, Chen et al. and Hua. et al. also showed sperm-borne tsRNA to be relevant for early embryo cleavage events and blastocyst formation[39,40], highlighting that both piRNA and tsRNA are highly relevant and interesting in the case of human reproduction.

One limitation in our analysis is the difficulty of examining the isolated effects of sperm sRNA on clinical outcome. Specifically, in analyses concerning rate of high-quality embryos, female factors influence the outcome. Additionally, within the studied population, positive correlations exist between females and males in both age and BMI. However, despite the female influence, we are still able to associate sperm-borne sRNA with embryonic quality, supporting their robustness in this system.

As embryo quality is one of the major limiting factors of modern IVF success, finding sperm-borne biomarkers capable of predicting embryo quality holds great potential. Couples often undergo several demanding IVF treatments before a successful pregnancy. Additionally, some couples are left without a diagnosis or explanation after failed IVF attempts, causing emotional distress. A more efficient selection of sperm can attain more high-quality embryos, alleviating the number of hormonal treatments needed, aiding in closing the inequality gap in reproductive medicine.

In conclusion, we identified different subpopulations of sRNA in sperm that exhibit distinct profiles in clinical reproductive measurements, such as sperm concentration, fertilisation rate, and the rate of high-quality embryos. Furthermore, our analysis identifies specific sRNA as highly promising biomarkers. These biomarkers are clinically applicable for the examination of sperm samples while providing insight into possible molecular mechanisms of male fertility and paternal contribution to embryonic development.

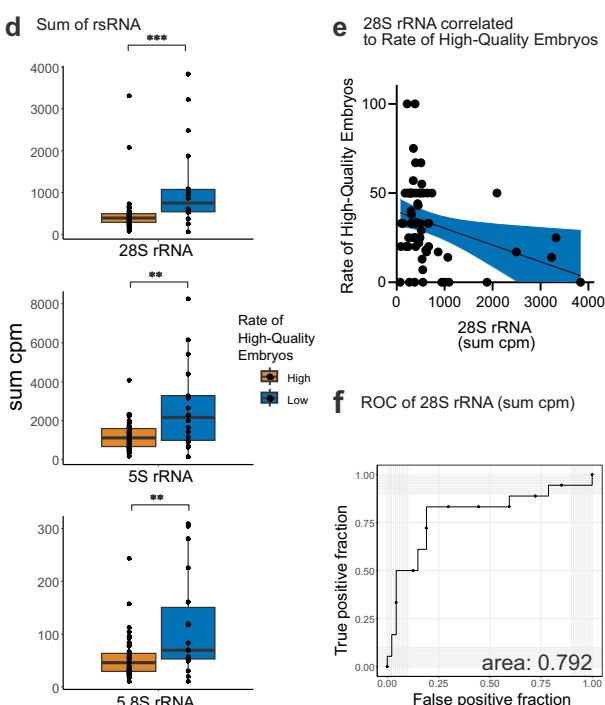

**miRNA Upregulated in High Rate of High-Quality Embryos**

**rsRNA Downregulated in High Rate of High-Quality Embryos**

**Fig. 5 | Sperm sRNA differs between samples of high and low rates of high-quality embryos. a** MiRNA with highest significance between high and low rate of high-quality embryos (≥/< 20%). Hsa-let-7g *P* val=0.000054, mir-320b/a *P* val=0.0067, hsa-mir-30d p-val=0.008. *P* values as shown in figure were produced by Wilcoxon nonparametric test (two-sided). **b** Linear regression comparing hsa-let-7g in sum cpm and rate of high-quality embryos (R² = 0.001, *P* value = 0.7903, equation= Y = -0,1167*X + 33,97). **c** ROC of sum cpm of hsa-let-7g to differentiate between high and low rate of high-quality embryos. **d** RsRNAs with highest significance between high and low rate of high-quality embryos. 28S rRNA=0.00018, 5S rRNA=0.0041, 5.8S rRNA=0.0096. *P* values as shown in figure were produced by Wilcoxon nonparametric test (two-sided). **e** Linear regression comparing 28 s rRNA in sum cpm and rate

of high-quality embryos (R² = 0.0837, *P* value = 0.0194, equation=Y = −0,009374* X + 39,65). **f** ROC of sum cpm of 28 s rRNA to differentiate between high and low rate of high-quality embryos. Orange=group of high rate of high-quality embryos, blue=group of low rate of high-quality embryos. *P* values for Wilcoxon nonparametric t-test, *≤0.05, **≤0.001, ***≤0.0001. Data is presented as sum of normalised sequences (cpm) for each sperm sample by indicated genomic origin. High rate of high-quality embryos n = 47, low rate of high-quality embryos n = 17. Line in box plot represents the median, hinges show the first and third quartiles and whiskers extend to the largest value unless values are above 1.5 times the inter-quartile range. In linear regression, coloured area represents 95% confidence interval.

## Methods

### Patients
Couples were recruited at the Centre for Reproductive Medicine at the University Hospital in Linköping, Sweden, on the day of ovum pick-up (OPU), the same day as semen sample collection and IVF fertilisation. As couples share an IVF record, written consent was obtained from both partners. The sex of participants was collected from their medical records. Couples did not receive any compensation for enroling.

Male participants had to meet the following criteria: the presence of mature sperm in the ejaculate and proficiency in Swedish. Males were excluded if they had a known malignant disease, a history of radiation in the pelvic area, previous chemotherapy, or a known genetic anomaly. Importantly, only males with over 600,000 leftover sperm were sequenced due to a restriction in input RNA level. Therefore, males with low sperm count requiring intra-cytoplasmic sperm injection could not be included.

Patients received standard care for IVF, where the only laboratory operation different from non-recruited couples was the retrieval of leftover sperm not used in the fertilisation process. Recruitment started on 18 November 2022 and ended on 8 June 2023.

### Sperm preparation
On the day of OPU, the semen samples were collected (at home or the clinic between 6:30–9:30) in sterile 50-mL non-spermotoxic poly-propylene Falcon tubes (Corning Science, Mexico) after recommendation of ejaculation every other day during hormonal stimulation

(8–14 days) and a maximum of 3 days abstinence. After liquefaction for 15–30 min at room temperature on a Nutating Mixer (VWR, Pennsylvania, USA), standard semen parameters were examined manually by embryologists. Parameters for concentration of sperm, concentration of progressively motile sperm and percentage of progressive motile and motile sperm in the ejaculate were evaluated according to WHO criteria[41]. Internal controls are performed every six months. Most samples were examined within 60 min post ejaculation and none later than 120 min.

Spermatozoa were prepared for IVF using PureSperm gradient (1.5 mL 80%/1.5 mL 40%) (Nidacon, Gothenburg, Sweden). Semen was layered on top of the gradient and centrifuged at 300 g for 20 min, followed by resuspension of the pellet (with PureSperm Wash; Nidacon). Subsequently, sperm was centrifuged at 500 g for 10 min, the pellet was resuspended, and the number and concentration of progressive motile sperm post gradient were estimated (Makler counting chamber, Cellvision, Heerhugowaard, Netherlands). From this post-gradient suspension, the sperm sample was prepared and diluted with G-IVF (VitroLife, Gothenburg, Sweden) for fertilisation. Leftover sperm from the sample was used for sRNA sequencing.

### Preparation of sperm for sRNA sequencing
After fertilisation on the day of OPU (between 13:30 to 15:30), leftover sperm was diluted with G-IVF (VitroLife) or concentrated (centrifugation on 500 g for 10 min and then fluid removal) to a concentration of 12 million sperm/mL. Aliquots of 50 µl (i.e., 600,000 sperm) were put in RNAse-free 2-mL Eppendorf tubes (Eppendorf, Germany), followed

by immediate freezing in liquid nitrogen. Frozen sperm was stored at −80 °C until extraction of sperm RNA.

## Small RNA library preparation and sequencing

RNA extraction was performed with miRNeasy Micro kit (Qiagen; 217084) according to the manufacturer's instructions. Prechilled Steel beads (stainless steel bead 5 mm, Qiagen; 69989) were added to frozen samples, followed by the addition of 1 mL of prechilled Qiazol (Qiagen). Samples were instantly processed in a Tissue Lyser LT (Qiagen) for 2 + 2 min at 40 oscillations/s upon defrosting. RNA concentration was determined with a Qubit microRNA assay kit (Q32880, Life Technologies) using Quantus Fluorometer (E6150; Promega, Madison, WI). RNA quality was controlled by BioAnalyzer 2100 (Agilent RNA 6000 Nano kit; 5067-1511), both by extracting RIN values and by visual inspection of the curves. The RIN values were low (2,3 -5,4), with no visible rRNA peaks indicative of pure sperm samples[42–44]. Library synthesis used 4 ng of RNA; if lower, full volume was used. Small RNA libraries were produced with NEBNext Multiplex SmallRNA Library Prep Kit for Illumina (E7300S, E7330S, E7580S; New England Biolabs), with the customisation of a dilution of primers 1:8. The 3´ adaptor ligation reaction was carried out at 16 °C overnight. Libraries were amplified for 15 cycles and cleaned using Agencourt AMPure XP (Beckman Coulter, Brea, CA). The quality of cDNA libraries was studied using BioAnalyzer 2100 with Agilent High Sensitivity DNA kit, 5067-4626. Amplified libraries were pooled in relation to peak size before size selection. Size selection was done using TBE gel (EC6265BOX; Invitrogen) of 130–165 nt length. The extraction of cDNA from gel was performed with Gel Breaker Tubes (3388-100, IST Engineering) by incubation with buffer included in the NEBNext kit and incubated on a shaker for 1 h at 37 °C, flash frozen for 15 min, and again incubated on a shaker. Gel debris was removed by Spin-X 0.45 μm centrifuge tubes (8162, Corning, Inc.). Precipitation was done using GlycoBlue (Invitrogen), 0.1 times the volume of Acetate 3 M (pH5.5) and three times the volume of 100% ethanol at −70 °C overnight. The final cDNA concentration was determined using a Quantus Fluorometer (E6150; Promega, Madison, WI) and QuantiFluor ONE ds DNA system (E4870, Promega). The quality of pooled and size selected cDNA libraries was studied again with BioAnalyzer and sequenced on NextSeq 550 with NextSeq 500/550 High Output kit version 2.5, 75 cycles (Illumina, San Diego, CA). All libraries passed Illumina's default quality control. Fastq files were produced by FASTQ generation v1.0.0 by Illumina.

## Bioinformatic analysis

Data analysis was performed within R version 4.3.2 with Seqpac version 1.2.0[1], mapping to human genome GRCh38. Any sequence that did not match the human genome with a maximum of 3 mismatches was discarded. Primary filters of 1 read in 90% of samples and a length of 16–75 nt were performed before normalisation by counts per million (cpm). Additional filtering of 1 cpm in 90% of samples was consecutively performed. Small RNA was annotated against ensemble ncRNA GRCh38.ncrna, piRNA from piRbase 2.0, protein coding from GRCh38 from NIH, and ribosomal RNA from RNAcentral, and searching for rRNA in Homo sapiens. Biotypes were annotated in a hierarchy as follows: rRNA, miRNA, mitochondrial RNA, tRNA, ribonucleoproteins (including SRPs, vaultRNA, snRNA, snoRNA, scaRNA, and Y-RNA), lncRNA, miscRNA, and piRNA. Biotypes were mapped with a mismatch of 1, where sequences failing to annotate to any of the previously defined databases were classified as "unannotated".

Data visualisation was performed with ggplot2 version 3.5.0. Differential expression analysis was performed with DESeq2[2] version 1.42.1, with the model accounting for flow cell (technical batch effect), as this is the major driver for variation in data when explored for with R package variancePartition version 1.32.5 and Principal Component Analysis performed with Seqpac (Supplementary Fig. 7a, b). In gestational age and size for gestational age, differential expression analysis

was performed as continuous variables. Wald test and local regression were used for differential expression.

Linear regression and normality check of cpm data were performed GraphPad Prism ver. 10.0.2. Results from tests for each RNA shown in boxplots may be found in Supplementary Data 12. Nonparametric Wilcoxon Rank Sum and Signed Rank Tests were two-sided and were performed with R function wilcox.test in stats version 4.3.2.

## Clinical outcomes

Clinical parameters from the couple's IVF records were retrieved, forming groups for analysis. Males with ≤16 million sperm/mL were classified as having low sperm concentration. This cut-off was based on WHO guidelines[41]. For Supplementary Fig. 2 and 7, ≥5 million progressively motile sperm in total amount after preparation with density gradient was considered as high sperm motility, <5 million sperm as low motility. For supplementary Fig. 7,[3] 30% motile sperm before density gradient was considered high motility percentage and <30% as low motility.

In line with The Vienna Consensus[4] for fertilisation rate (n fertilised oocytes/n oocytes retrieved) and benchmark values for the Centre of Reproductive Medicine Linkoping, Sweden, the ratio of ≥70% was set as a high fertilisation rate.

Embryologists examined the embryos through microscope with criteria mentioned below. For cleavage stage embryos (days 2 and 3), the guidelines from the Istanbul consensus were used[45]. At day 2, High-quality embryos were defined as embryos with 4–6 evenly sized blastomeres with a maximum of 20% fragmentation and a smooth cytoplasmic appearance. At day 3, embryos with 6–10 evenly sized blastomeres with a maximum of 20% fragmentation and a smooth cytoplasmic appearance were defined as high-quality. For blastocyst stage embryos (day 5), embryos were graded according to Gardner. High-quality embryos were considered to have a grade of 3BB or higher[46]. Couples with a high rate of high-quality embryos had a ratio of ≥20% (n high-quality embryos/ n fertilised oocytes*100).

In Supplementary Fig. 6, size of gestational age is calculated by neonatal weight charts resulting in z-scores corresponding to centiles[47]. Small for gestational age (SGA) is defined as <10th centile. Average for gestational age (AGA) is defined as 10-90th centile. Large for gestational age (LGA) is defined as <90th centile. These calculations were performed with web tool from fetalmedicine.com.

## Reporting summary

Further information on research design is available in the Nature Portfolio Reporting Summary linked to this article.

## Data availability

This study involves human participants and is covered by an ethical permit granted by the Swedish Ethics Board under number 2022-00244-01. The sequencing data are available under restricted access for ethical permit and data privacy law reasons, access can be obtained by extension of ethical permit by contacting anita.ost@liu.se. Response time after mail contact is expected within weeks and data will be available as long as needed for purpose specified in extended ethical permit. The processed data generated in this study are provided in the Supplementary material.

## Code availability

The code used in this study is available at github.com/sign-eisacson/KIPF.

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

## Acknowledgements

We wish to thank the personnel at the Centre for Reproductive Medicine at the University Hospital at Region Östergötland for their patience and assistance while setting up this study in parallel with their standard work. We thank collaborators and colleagues for their evaluation and feedback on this work. We acknowledge the Core Facility at the Faculty of Medicine and Health Sciences, Linköping University for providing assistance in sequencing. Financial support was provided by The Swedish Research Council (2015-03141, A.Ö), The Swedish Research Council (2020-00577, A.Ö), Ragnar Soderberg's foundation, Knut and Alice Wallenberg foundation (2015.0165, A.Ö), ALF Grants Region Östergötland (RÖ-995139, A.Ö), ALF Grants Region Östergötland (RÖ-975378, A.Ö).

## Author contributions

S.I., K.K., U.K., S.L., S.Z., A.A., and A.Ö conceived the project. S.I. performed bioinformatic analysis, data curation, and manuscript writing. K.K. performed sample collection, data curation, and manuscript writing. U.K. performed sequencing. A.A. conceptualised and performed manuscript revision. S.Z. and S.L. supervised clinical sampling of material and data and performed manuscript revision. S.L. and K.K. recruited patients. A.Ö. conceptualised, provided funding, performed manuscript writing, and supervised the project.

## Funding

## Competing interests

The authors declare no competing interests.
