## [Peer Review file · Nature Communications]

Small RNA in sperm – paternal contributions to human embryo development

Corresponding Author: Professor Anita Öst

Version 0:

Reviewer comments:

Reviewer #1

(Remarks to the Author)

The study from Isacson et al. reports a characterization of human sperm small non-coding RNAs (sRNAs) and their potential association with sperm parameters and IVF outcomes. The authors conclude that sperm sRNAs could be valuable biomarkers for sperm concentration and IVF embryo quality. Despite the relatively small cohort size, the study is well designed, well written and timely.

Here my general comments:

1. As sperm sRNAs are gaining more and more attention as predictors/reporters of male pre-conceptional health, to be valuable biomarkers of sperm and IVF outcomes, potential confounding factors of general male health (BMI, lifestyle, glucose and lipid profiles) should be excluded/accounted in the analysis.
2. The authors isolate two associations between sperm tRNAs (and their fragments?) with sperm concentration and miRNAs with embryo quality. Of the two, sperm tRNAs seem better predictors than miRNAs (according to the correlation's analysis and ROC curves). What about associations with other pregnancy outcomes: placentation and placental function, fetal developmental trajectories, incidence of pregnancy complications, gestational age at delivery?
3. The authors also report that 28/65 successful fertilizations resulted in live children. It would be interesting to associate sperm sRNA profiles with birthweight.

All the Best,
Raffaele Teperino

(Remarks on code availability)

The codes are well written and easily reproducible.

Reviewer #2

(Remarks to the Author)

In this manuscript, the authors detail the association between sperm sRNA and fertility factors in men seeking fertility treatment. The authors have analyzed 70 single sperm samples from couples undergoing IVF treatment, sequenced the sRNA and then modeled these expression levels against the sperm factors of motility, fertilization, and embryo quality. However, no endpoints of pregnancy success or fetal outcomes were included. These are interesting data that largely support previous studies that have identified similar RNAs associated with fertility outcomes. The addition of mitochondrial and Y-RNAs here is a novel addition to the field. The work overall is solid, pending additional statistical and method details, but the level of novelty and biological relevance is less clear. To this point, the statement of the main findings in the abstract, "We demonstrate that distinct sets of sperm-borne sRNA levels are important for each stage", is not demonstrated in these studies as the authors have not tested for any causal outcomes, only modeled associations. Below are concerns with some of the methods and statistics that could be improved with more details.

1. A major question for the data is the quality of the RNA extracted. The methods do not provide detail as to how the RNA quality was examined as there was no mention of RIN scores or quality control steps. Was only concentration of RNA

examined?

2. In addition, the authors do not provide the raw RNA data for reader examination stating subject privacy. However, these are deidentified samples it is not clear why there would be a concern with subject privacy. For evaluation of data quality, all raw files should be provided for RNA sequences.

3. It is surprising that no mention of pregnancy success or other outcomes was reported as this would greatly improve the paper's impact by providing information as to biological relevance of the associations between the sRNA expression levels and IVF success.

4. While sperm motility is a major factor examined, there is limited method details provided. What component of motility was measured? What was the software used to assess velocity? How did the authors define up or down regulated motility if they are using a percent motile or a motile count? These are essential details to interpret the authors' results.

5. While batch effects were controlled for in sequencing, there are a number of other factors that need corrections in the statistical comparisons including time of day of collection and how samples were collected (lab or at home), as well as subject factors of BMI, age, and FSH dose. The variance of these factors from Table 1 suggests a high potential for bias of results with these subject factors.

6. Statistical analyses need additional details: The authors are using a lot of nonparametric testing, and it would be helpful if they stated why. A paragraph in the results discussing the authors' decisions for statistical methods would be helpful, including reporting the degrees of freedom.

7. Fig 4b, single data point seems like a clear outlier in the linear regression analysis. Was this statistically probed for exclusion?

8. Fig 2 shows profound effects of tRNAs and piRNAs, but these are not discussed.

9. For embryo quality, the blastomere fragmentation analysis is not explained.

(Remarks on code availability)

Version 1:

Reviewer comments:

Reviewer #1

(Remarks to the Author)

Dear Signe, Anita and all co-authors,

thanks a lot for taking time to fully address my comments, and good luck with the new study on male health and life-style intervention before IVF. I would love to see the results.

Congratulations and all the Best,
Raffaele Teperino

(Remarks on code availability)

I assessed the codes in the first round of revisions and they are well written and extensively explained to be easily reproduced.

Reviewer #2

(Remarks to the Author)

The authors have done an impressive job addressing all of my comments and the manuscript is impactful and exciting.

I have no further comments. Great job to the authors!

(Remarks on code availability)

I cannot review the authors raw data for the sperm sncRNA as they are not allowed to make it public.

Response to Reviewer #1:

Thank you for the feedback. In response to the two reviews, we have included several new analyses that strengthen the manuscript. Please find below a synopsis of the major additions, followed by the point-by-point response to your comments.

As suggested by both reviewers, we have analysed sperm-borne sRNA in relation to live birth, gestational age, and size for gestational age. We are happy for this suggestion as we identified interesting sRNA both in relation to live birth and size for gestational age. More specifically, regarding live birth, we identified significant difference in a few sRNA from U6. Moreover, we found an association between size for gestational age and certain sperm-borne tsRNA. This data is presented as a new supplementary figure (Supplementary Figure 6). We have additionally updated Table 1 to match the new Supplementary Figure 6.

Moreover, we have added analysis and text to explain the rationale for choosing covariates for the DE-seq analysis in a new figure (Supplementary Figure 7). We have also clarified methodological details in text.

Lastly, we have added a more in-depth analysis regarding the piRNA and tsRNA associated with fertilisation and the data is presented in a new main figure (Figure 4). Intriguingly, the sequences that associate with fertilisation all originate from the same genomic region, suggesting this region to be important for the sperm's capabilities of fertilising the oocyte. In addition, we have added text in the result section to match the new data. Changes in the text are shown with tracked changes and row numbers are referred to with tracked changes on.

Reviewer #1 (Remarks to the Author):

The study from Isacson et al. reports a characterization of human sperm small non-coding RNAs (sRNAs) and their potential association with sperm parameters and IVF outcomes. The authors conclude that sperm sRNAs could be valuable biomarkers for sperm concentration and IVF embryo quality. Despite the relatively small cohort size, the study is well designed, well written and timely.

Thank you, Raffaele, for your valuable insights on our manuscript.

Here my general comments:

1. As sperm sRNAs are gaining more and more attention as predictors/reporters of male pre-conceptual health, to be valuable biomarkers of sperm and IVF outcomes, potential confounding factors of general male health (BMI, lifestyle, glucose and lipid profiles) should be excluded/accounted in the analysis.

Thank you for your input. We agree with you that it is interesting to see if male health parameters shift the sperm sRNA profile. We are right now recruiting patients to a randomized controlled trial with dietary intervention before IVF at the Center of Reproductive Medicine in Linköping. We hope this will answer the question if diet shifts the sperm sRNA profile, and if the successes rate is improved. It will take 1-2 years to

complete, so it is too early to include in this story. Thus, we don't have any lifestyle data to include in this manuscript. Neither do we have samples to test glucose and lipid profile.

Regarding potential confounding factors in the differential expression analysis, we used two strategies to determine what factors to include. First, we performed a Variance Partition Analysis, that identified the flow cell to explain the most variance. Second, we used PCA plots to see if male age and BMI would influence the clustering of data. This analysis supported the result from the Variance Partition analysis by not showing any clustering by age or BMI. Thus, we made the decision to have a model with only the flow cell as a covariate.

We have now added text regarding the rationale for choosing confounding factors in the method on line 375-377, and added the results from the Variance Partition analysis and PCA plots as Supplementary Figure 7. The text reads as follows:

“Differential expression analysis was performed with DESeq2² version 1.42.1, with the model accounting for flow cell (technical batch effect), as this is the major driver for variation in data when explored for with R package variancePartition version 1.32.5 and Principal Component Analysis (Supplementary Fig. 7 a,b).”

In addition, we would like to point out that while the differential expression analysis was used to identify interesting sRNA, our linear regression, t-test and clinically relevant ROC curves (Figure 3 and 5), strengthens our findings from the negative binomial model of DESeq2.

2. The authors isolate two associations between sperm tRNAs (and their fragments?) with sperm concentration and miRNAs with embryo quality. Of the two, sperm tRNAs seem better predictors than miRNAs (according to the correlation's analysis and ROC curves). What about associations with other pregnancy outcomes: placentation and placental function, fetal developmental trajectories, incidence of pregnancy complications, gestational age at delivery?

Yes, you are right. The mitochondrial tRNA-fragment from TS1-Ser1 have a positive correlation with sperm concentration, and a ROC AUC score of 0.891. This is better than the ROC scores we get for miRNA and embryo quality. Still, a ROC AUC score above 0.8 is considered good, and for embryo quality and let 7-g, we get 0.812.

Regarding the pregnancy outcomes you mention, we have some of the data you ask for but not all. We don't have any data about placentation and placental function. Of the subjects studied, we had 7 miscarriages. Of the 28 children born we know of these complications: 3 were born prematurely (but all in week 35 or above) and 1 had hypospadias. Of the babies born, two were small for gestational age and five were large, and the rest (21) were average for gestational age. These populations are too small to make any meaningful statistical comparisons of.

We have data concerning live birth, gestational age and size for gestational age, and have added analysis of sperm sRNA in relation to these into the manuscript (Supplementary Figure 6). We find sperm-borne snRNA from U6 to be related to chance of live birth, and certain tsRNAs related to size for gestational age. We have also added text in result section on line 160-180 and discussion on line 248-253.

3. The authors also report that 28/65 successful fertilizations resulted in live children. It would be interesting to associate sperm sRNA profiles with birthweight.

Please see the response on point 2.

All the Best,
Raffaele Teperino

Response to Reviewer #2:

Thank you for the feedback. In response to the two reviews, we have included several new analyses that strengthen the manuscript. Please find below a synopsis of the major additions, followed by the point-by-point response to your comments.

As suggested by both reviewers, we have analysed sperm-borne sRNA in relation to live birth, gestational age, and size for gestational age. We are happy for this suggestion as we identified interesting sRNA both in relation to live birth and size for gestational age. More specifically, regarding live birth, we identified significant difference in a few sRNA from U6. Moreover, we found an association between size for gestational age and certain sperm-borne tsRNA. This data is presented as a new supplementary figure (Supplementary Figure 6). We have additionally updated Table 1 to match the new Supplementary Figure 6.

Moreover, we have added analysis and text to explain the rationale for choosing covariates for the DE-seq analysis in a new figure (Supplementary Figure 7). We have also clarified methodological details in text.

Lastly, we have added a more in-depth analysis regarding the piRNA and tsRNA associated with fertilisation and the data is presented in a new main figure (Figure 4). Intriguingly, the sequences that associate with fertilisation all originate from the same genomic region, suggesting this region to be important for the sperm's capabilities of fertilising the oocyte. In addition, we have added text in the result section to match the new data. Changes in the text are shown with tracked changes and row numbers are referred to with tracked changes on.

In this manuscript, the authors detail the association between sperm sRNA and fertility factors in men seeking fertility treatment. The authors have analyzed 70 single sperm samples from couples undergoing IVF treatment, sequenced the sRNA and then modeled these expression levels against the sperm factors of motility, fertilization, and embryo quality. However, no endpoints of pregnancy success or fetal outcomes were included. These are interesting data that largely support previous studies that have identified similar RNAs associated with fertility outcomes. The addition of mitochondrial and Y-RNAs here is a novel addition to the field. The work overall is solid, pending additional statistical and method details, but the level of novelty

and biological relevance is less clear. To this point, the statement of the main findings in the abstract, “We demonstrate that distinct sets of sperm-borne sRNA levels are important for each stage”, is not demonstrated in these studies as the authors have not tested for any causal outcomes, only modeled associations. Below are concerns with some of the methods and statistics that could be improved with more details.

Thank you for your summary and your appreciation of our work. We have now, as you suggested, analysed sperm sRNA in relationship to IVF success rate, live birth, gestational age and size for gestational age (Supplementary Figure 6). These findings are discussed more under point 3 below. Moreover, we have removed the sentence “We demonstrate that distinct sets of sperm-borne sRNA levels are important for each stage” in the abstract. Additionally, we have made the choice of statistical analysis clearer in text and added further information on data normality (Supplementary Table 12), as well as clarification concerning covariates in differential expression analysis (Supplementary Figure 7).

1. A major question for the data is the quality of the RNA extracted. The methods do not provide detail as to how the RNA quality was examined as there was no mention of RIN scores or quality control steps. Was only concentration of RNA examined?

We agree that good RNA quality is crucial, and we have indeed tested the RNA quality with Bioanalyzer kit RNA 6000. This information was missing in the method, and we have now added the following text on line 335-338:

“RNA quality was controlled by BioAnalyzer 2100 (Agilent RNA 6000 Nano kit; 5067-1511), both by extracting RIN values and by visual inspection of the curves. The RIN values were low (2,3 -5,4), with no visible rRNA peaks indicative of pure sperm samples¹⁻³.”

In addition to RNA quality, cDNA was tested with Bioanalyzer and Quantus as a quality control before sequencing, as described on line 353-356.

2. In addition, the authors do not provide the raw RNA data for reader examination stating subject privacy. However, these are deidentified samples it is not clear why there would be a concern with subject privacy. For evaluation of data quality, all raw files should be provided for RNA sequences.

We appreciate the concern, as we wish to share as much data as we can to our scientific community. However, under the ethical permit where samples were collected, we are not able to publicly distribute raw data. We therefore present filtered data in Supplementary Table 1, where both normalised counts and sequences are presented. As now clarified in the Data availability section on line 413-418, it is possible to share files after extension of ethical permit, which may be performed together with Anita Öst and the Swedish Ethical Review Authority. The text now reads:

“This study involves human participants and is covered by an ethical permit granted by the Swedish Ethics Board under number 2022-00244-01. Due to ethical considerations of the participants, raw sequencing data produced for this manuscript will not be

publicly available. Normalised and filtered data are available in the supplementary material. Regarding safe sharing of raw data via extension of ethical permit, please contact anita.ost@liu.se.”

3. It is surprising that no mention of pregnancy success or other outcomes was reported as this would greatly improve the paper’s impact by providing information as to biological relevance of the associations between the sRNA expression levels and IVF success.

We agree with the reviewer, and we have now added a new figure describing this (Supplementary Figure 6) and corresponding text in result section on line 160-180 and discussion on line 248-253. This analysis revealed sperm-borne snRNA U6 to correlate with chance of live birth as well as a few tsRNA to correlate with size for gestational age. This find is interesting since experiments in mice has identified sperm-borne tsRNA to be sensitive to dietary changes and to influence the metabolism of next generation^{4,5}. It is too early to say if similar mechanisms are at work in humans, and further work is clearly needed.

4. While sperm motility is a major factor examined, there is limited method details provided. What component of motility was measured? What was the software used to assess velocity? How did the authors define up or down regulated motility if they are using a percent motile or a motile count? These are essential details to interpret the authors’ results.

Thank you for bringing this to our attention. The Center of Reproductive Medicine evaluates sperm quality parameters according to the WHO criteria. The embryologists evaluate sperm manually regarding sperm concentration, progressive motile sperm concentration, progressive motile and motile sperm in percentage before preparation. Total number and concentration of progressive motile sperm is evaluated again after density gradient. When performing the analysis shown in Supplementary Figure 2, the groups compared are >/< 5 million progressive sperm after preparation. This is now clarified in the results on line 84-85 and methods section on line 307-310, 316 and 388-392. We have also clarified sperm parameters in Table 1, Supplementary Table 2, 4 and 6 accordingly.

5. While batch effects were controlled for in sequencing, there are a number of other factors that need corrections in the statistical comparisons including time of day of collection and how samples were collected (lab or at home), as well as subject factors of BMI, age, and FSH dose. The variance of these factors from Table 1 suggests a high potential for bias of results with these subject factors.

We agree that it is of importance to understand which factors may influence the outcome of the analysis and have now added additional information about how we decided on covariates in the differential expression analysis.

Concerning time of day of collection, the couples come to the clinic for oocyte retrieval either at 07:00 or at 9:00, where oocytes are collected around 8:30 and 10:00. Male ejaculates are collected around 6:30 to 9:30 (if provided at home), and all treatments undergo fertilisation at around 13:30. The leftover sperm is then prepared and put in liquid nitrogen between 13:30-15:30 depending on the patient flow for the day. Thus,

there is little variation between samples in timepoints of preparation. This has been clarified in the methods section on line 302-310 and 322. We have also clarified text regarding abstinence days before semen sample collection on line 305.

To decide which covariates to include in the DESeq2-analysis, we first performed a Variance Partition Analysis. The major factor identified to be of relevance was the flow cell (batch effect). We see little variance explained by where samples were collected, BMI, age and FSH dose. Next, to verify the results from the Variance Partition Analysis, we used PCA to see if age, BMI or FSH per oocyte would influence the overall dimensions of the data. In agreement with the Variance Partition Analysis, we see little effect of the mentioned factors on the dimensions of the sRNA data. These two analyses are now presented in a new figure (Supplementary Figure 7).

To further confirm lack of association between female/male age, BMI and FSH dose per oocyte and *let-7g*, *28S rRNA*, *MT-TS1-Ser1* and *RNY4*, we also tested this with correlation. We found no significant correlation of four RNAs of interest to age, BMI for either male or female, nor for FSH per oocyte dose (shown in table below).

	R ²					p -value				
	Age M	Age F	BMI M	BMI F	FSH/ooc	Age M	Age F	BMI M	BMI F	FSH/ooc
let-7g	-0,130	-0,065	-0,192	-0,117	-0,099	0,282	0,593	0,110	0,336	0,414
28S rRNA	-0,077	-0,116	0,015	0,012	0,126	0,541	0,356	0,903	0,924	0,318
MT-TS1-Ser1	0,000	0,051	-0,079	-0,076	-0,013	0,998	0,676	0,515	0,533	0,913
RNY4	-0,157	-0,192	-0,080	0,037	0,009	0,195	0,112	0,508	0,764	0,940

Thus, we decided to only use the flow cell as a covariate in the DESeq2 analysis. We believe this decision to be sound as our linear regression, t-test and clinically relevant ROC analysis (Figure 3 and 5), validates our findings outside of the negative binomial model of DESeq2. The rationale behind our choice of covariates is now described in the methods section on line 375-378. It reads:

“Differential expression analysis was performed with DESeq2² version 1.42.1, with the model accounting for flow cell (technical batch effect), as this is the major driver for variation in data when explored for with R package variancePartition version 1.32.5 and Principal Component Analysis (Supplementary Fig. 7 a,b).”

Still, and as you state, there is variance in the clinical parameters between the groups high vs low rate of high-quality embryos. As seen in Supplementary Table 6, we see a difference especially in female age between these groups. Thus, we decided to directly test if our main findings could be affected by female age. We age-matched participants from high and low rate of high-quality embryos with nearest-neighbour matching with R package MatchIt and then repeated prior analysis comparing sum cpm of high and low groups with Wilcoxon t-test for the miRNA 320a, *let-7g* and 30d. This analysis still show significant differences for all three miRNA, supporting that our original analysis and findings is not due to differences in the female age.

6. Statistical analyses need additional details: The authors are using a lot of nonparametric testing, and it would be helpful if they stated why. A paragraph in the results discussing the authors' decisions for statistical methods would be helpful, including reporting the degrees of freedom.

Thank you for highlighting the need to discuss the choice of statistical analysis. As this data is count-based, much of the data is not normally distributed and that is why we have performed nonparametric testing with Wilcoxon. We have now added further explanation of the statistical analysis performed in the methods section on line 381-384, as well as normality test results in Supplementary Table 12. Concerning degrees of freedom, this is not commonly described in the case of non-parametric tests. We have however clarified figure legends, especially focusing on the n-values for each group, to make the grouping underlying the Wilcoxon more evident. The text on line 381-384 reads:

“When performing nominal statistics on cpm, data was checked for normality with GraphPad Prism ver. 10.0.2. Results from tests for each RNA shown in boxplots may be found in Supplementary Table 12. Non-parametric Wilcoxon Rank Sum and Signed Rank Tests was performed with R function wilcox.test in stats version 4.3.2.”

7. Fig 4b, single data point seems like a clear outlier in the linear regression analysis. Was this statistically probed for exclusion?

This sample (SK16) was behaving accordingly in all other analysis, as can be observed by the lack of outliers in all other boxplots presented both in supplementary and main figures. Thus, we believe there is little reason to exclude this sample based on expression of one miRNA. We however agree that this sample is a likely outlier, as its cpm levels are twelve times the mean of all other samples. With this sample excluded from that particular linear regression, we indeed receive a significant result. We now present one version of the graph in Supplementary Figure 4 c without this sample and discuss this on lines 144-146. It reads:

“However, exclusion of one sample with an expression 12 times above mean, resulted in a significant regression (Supplementary Fig. 4c).”

8. Fig 2 shows profound effects of tRNAs and piRNAs, but these are not discussed.

Thank you for the comment. We agree that these are interesting findings, and we have now added this analysis as a new main figure (Figure 4). In short, sequences found in the DESeq2-analysis to be significantly downregulated in sperm with high fertilisation rate (Fig 2b), all come from the same genomic region. This region has overlapping sequences annotated as piRNA and tRNA. We discuss these findings in the results section on line 121-131 and in the discussion on line 246-257. It reads:

“We identified 34 sequences of sperm-borne sRNA to be associated with low fertilisation rate (Figure 2b, Supplementary Table 5). A closer analysis of these sequences revealed that they all originate from the same genomic region (Figure 4a). Sequences from this region can be annotated as both tRNA and piRNA depending on database. It is therefore not possible to precisely classify the differentially expressed sequences found here. Comparing the sum of normalised sequences in samples with high or low fertilisation rate did not show a statistically significant difference (Figure 4b, c). In addition, ROC analysis showed an AUC of 0.5817 (Figure 4d). In all, our data suggests that sequences from this specified region may be relevant for the sperm’s capability of fertilisation, but more data would be needed to strengthen this finding.”

And

“Regarding fertilisation rate, we find sequences of piRNA or tsRNA originating from the same locus to be differentially expressed, suggesting a potential intriguing role for this locus in male fertility. We also find other sperm-borne tsRNA, namely from Arg-CCT-3-1, iMet-CAT-1-1, Glu-TTC-14-1 and Lys-CTT-2-1 to have a trend of negative correlation with size for gestational age. In mouse, sperm-borne tsRNA have been related to offspring body weight⁶. However, in a human setting, these findings need to be further explored in larger cohorts since foetal growth as well as fertilisation is heavily dependent on female factors before and after conception. Nonetheless, Oluwayiose et al.⁷ identified two sperm-borne piRNAs to be relevant for the chance of live birth. Additionally, Chen et al. and Hua. et al. also showed sperm-borne tsRNA to be relevant for early embryo cleavage events and blastocyst formation^{8,9}, highlighting that both piRNA and tsRNA are highly relevant and interesting in the case of human reproduction.”

9. For embryo quality, the blastomere fragmentation analysis is not explained.

The blastomere fragmentation analysis is performed via visual inspection by experienced embryologists following the guidelines from the Istanbul consensus and Gardner classification as described in the methods section. Internal controls at the

Center of Reproductive Medicine are done twice a year to ensure high quality of gradings. We have clarified this in the methods section on line 396-398. It reads:

“Embryologists examined the embryos through microscope with criteria mentioned below. For cleavage stage embryos (days 2 and 3), the guidelines from the Istanbul consensus were used¹⁰”

References

1. Bianchi E, Stermer A, Boekelheide K, et al. High-quality human and rat spermatozoal RNA isolation for functional genomic studies. *Andrology* [Internet] 2018 [cited 2025 Apr 29];6(2):374. Available from: <https://pmc.ncbi.nlm.nih.gov/articles/PMC6309170/>
2. Roszkowski M, Mansuy IM. High Efficiency RNA Extraction From Sperm Cells Using Guanidinium Thiocyanate Supplemented With Tris(2-Carboxyethyl)Phosphine. *Front Cell Dev Biol* [Internet] 2021 [cited 2025 Apr 29];9. Available from: <https://pubmed.ncbi.nlm.nih.gov/33968930/>
3. Gòdia M, Mayer FQ, Nafissi J, et al. A technical assessment of the porcine ejaculated spermatozoa for a sperm-specific RNA-seq analysis. *Syst Biol Reprod Med* [Internet] 2018 [cited 2025 Apr 29];64(4):291–303. Available from: <https://www.tandfonline.com/doi/pdf/10.1080/19396368.2018.1464610>
4. Chen Q, Yan W, Duan E. Epigenetic inheritance of acquired traits through sperm RNAs and sperm RNA modifications. *Nat Rev Genet* [Internet] 2016 [cited 2018 Mar 15];17(12):733–43. Available from: <http://www.ncbi.nlm.nih.gov/pubmed/27694809>
5. Sharma U, Conine CC, Shea JM, et al. Biogenesis and function of tRNA fragments during sperm maturation and fertilization in mammals. *Science (80-)* [Internet] 2016 [cited 2019 May 13];351(6271):391–6. Available from: <http://www.ncbi.nlm.nih.gov/pubmed/26721685>
6. Zhang Y, Ren L, Sun X, et al. Angiogenin mediates paternal inflammation-induced metabolic disorders in offspring through sperm tsRNAs. *Nat Commun* 2021 121 [Internet] 2021 [cited 2025 May 9];12(1):1–11. Available from: <https://www.nature.com/articles/s41467-021-26909-1>
7. Oluwayiose OA, Houle E, Whitcomb BW, et al. Urinary phthalate metabolites and small non-coding RNAs from seminal plasma extracellular vesicles among men undergoing infertility treatment. *Environ Pollut* [Internet] 2023 [cited 2024 Jul 11];329. Available from: <https://pubmed.ncbi.nlm.nih.gov/37003585/>
8. Chen X, Sun Q, Zheng Y, et al. Human sperm tsRNA as potential biomarker and therapy target for male fertility. *Reproduction* [Internet] 2021 [cited 2025 Feb 6];161(2):111–22. Available from: <https://pubmed.ncbi.nlm.nih.gov/33434159/>
9. Hua M, Liu W, Chen Y, et al. Identification of small non-coding RNAs as sperm quality biomarkers for in vitro fertilization [Internet]. Nature Publishing Group; 2019 [cited 2020 Jan 24]. Available from: </pmc/articles/PMC6453904/>
10. Balaban B, Brison D, Calderón G, et al. The Istanbul consensus workshop on embryo assessment: proceedings of an expert meeting. *Hum Reprod* [Internet] 2011 [cited 2024 Nov 1];26(6):1270–83. Available from: <https://pubmed.ncbi.nlm.nih.gov/21502182/>